# Long-Term Creep Compliance of Wood Polymer Composites: Using Untreated Wood Fibers as a Filler in Recycled and Neat Polypropylene Matrix

**DOI:** 10.3390/polym14132539

**Published:** 2022-06-22

**Authors:** Marko Bek, Alexandra Aulova, Klementina Pušnik Črešnar, Sebastjan Matkovič, Mitjan Kalin, Lidija Slemenik Perše

**Affiliations:** 1Faculty of Mechanical Engineering, University of Ljubljana, 1000 Ljubljana, Slovenia; marko.bek@fs.uni-lj.si (M.B.); sebastjan.matkovic@fs.uni-lj.si (S.M.); mitjan.kalin@fs.uni-lj.si (M.K.); 2Department of Materials and Manufacturing Technology, Chalmers University of Technology, 41296 Gothenburg, Sweden; aulova@chalmers.se; 3Faculty of Mechanical Engineering, University of Maribor, 2000 Maribor, Slovenia; klementina.pusnik@um.si

**Keywords:** wood–polymer composites, creep, durability, wood, recycling

## Abstract

Neat (NPP) and recycled (RPP) polypropylene matrix materials were used to prepare wood–polymer composites with untreated wood fibers up to 40 wt.%. Long-term creep properties obtained through the time-temperature superposition showed superior creep resistance of composites with NPP matrix. In part, this is attributed to their higher crystallinity and better interfacial adhesion caused by the formation of a transcrystalline layer. This difference resulted in up to 25% creep compliance reduction of composites with NPP matrix compared to composites with recycled (RPP) polypropylene matrix, which does not form a transcrystalline layer between the fibers and polymer matrix. Despite the overall inferior creep performance of composites with RPP matrix, from the 20 wt.% on, the creep compliance is comparable and even surpasses the creep performance of unfilled NPP matrix and can be a promising way to promote sustainability.

## 1. Introduction

The use of alternative and sustainable polymeric materials is increasing in the light of growing environmental issues and more restrictive legislation. One practical alternative to petroleum-based polymers are wood (thermoplastic) polymer composites (WPC). Using wood fibers in WPC offers many advantages such as low density, wood-like appearance, solid mechanical properties, biodegradability, renewability, easier processing, lower cost, and lower wear of processing equipment [1,2]. The sustainability of WPC materials can be even further increased if a recycled polymeric (or even biodegradable) matrix and waste wood are used. WPC materials have long been used for outdoor decking and other decorative structures. They have also found their use in the packing, furniture, and automotive industry [1,2]. However, WPC materials are mostly lightly loaded and are not used as structural elements [3].

More extensive use of WPC materials in demanding environmental and loading conditions is limited by the sensitivity of materials to moisture/swelling, temperature, and UV [1]. Additionally, the mechanical properties of WPC change with time, which can be an issue for long-term use in load-bearing applications. One of the main limiting factors for WPC materials is their incompatibility between the hydrophobic polymeric matrix and hydrophilic wood fibers/particles. Furthermore, polarity difference leads to bad interfacial adhesion, which causes poor thermo-mechanical properties [2]. Hence, the interface plays a crucial role in dictating the thermo-mechanical properties of composites. The issues with the interface and an overview of different bonding mechanisms, modifications for better interfacial adhesion, and characterization techniques to determine interfacial properties are given in three excellent and extensive papers [4,5,6].

Among the different ways to increase the compatibility between wood and polymeric matrix, there are several physical and chemical techniques [7]. Physical treatment includes high-temperature treatments, UV and gamma radiation, and cold plasma or corona treatment [5,7]. Chemical treatments include chemical treatment or impregnation of fibers [5,7,8] or using coupling agents where maleic anhydride grafted polyolefins are often used [9,10,11,12]. However, other coupling agents are also used (silane, isocyanate, ionomers) [7,13,14]. The benefits of increased interfacial adhesions are shown through improved mechanical properties [8,9,11,15,16]. The drawbacks of all additional treatments are related to higher costs [17]. Furthermore, they can have limited time effectiveness and can be environmentally problematic [8]. Additionally, treatments can decrease biodegradability (when the biodegradable polymeric matrix is used) and make the recycling process more challenging.

The viscoelastic nature of the polymeric matrix and, to a lesser extent, also of wood fibers causes the WPC materials to creep. Thus, the creep performance of the WPC materials is partially determined by the underlying creep properties of matrix materials. Several different polymeric matrix materials were already investigated such as polyester [8], polyvinyl acetate (PVC) [18,19,20], polyethylene (PE) [3,11,13,14,15,21,22,23] and polypropylene (PP) [3,9,10,19,24]. However, biodegradable and bio-based polymeric matrices were also investigated with respect to WPC [17,25,26,27,28].

In addition to the polymeric matrix, the wood fibers’ size and aspect ratio also play a role in creep performance [29]. Generally, the shorter fibers result in better creep performance as the specific surface area of particles is higher, and better interaction with the polymeric matrix can be achieved.

Concerning the wood fiber concentration, typically, up to 50 wt.% of wood fibers are used, and it is known that with the addition of wood fibers, the creep is decreased as the movement of polymer molecules is hindered by the presence of relatively rigid wood fibers, and also there is less polymer material that creeps [30]. This was observed by many researchers across the different fiber sizes and types, different matrices, and coupling agents [11,20,24,26,27], providing that a good dispersion and wetting of fibers is achieved [10]. Furthermore, improved creep behavior is also shown if the interfacial adhesion is enhanced, as this enables more load to be transferred from the polymeric matrix to the wood fibers [8,9,11,16,21,28,31].

The long-term creep properties of WPC materials can be determined using the time-temperature superposition (tTSP) principle. It has been shown that tTSP can be applied to WPC materials [9,12,20,23,27], and the validity of the predicted long-term creep behavior was confirmed by several authors [10,19,20].

Although many researchers investigated the creep properties of WPC materials, less is known about the long-term creep properties of WPC materials where wood fibers are used only as a filler, where there is no chemical or mechanical (pre) treatment of fibers. While such composites do not provide the same creep performance and other (thermo) mechanical properties as WPC materials with additives that enhance interfacial adhesion, wood as a filler can still increase creep performance compared to the neat polymeric matrix. Moreover, using untreated fibers can be helpful for the formation of transcrystalline morphology in the presence of fibers. Generally, the transcrystalline layer forms more easily on the unmodified wood than on chemically treated wood [32]. The transcrystalline phase around fibers has a very positive effect on the interfacial adhesion, load transfer, and in turn, on (thermo) mechanical properties [33,34,35]. Furthermore, there could also be economical and environmental benefits (ease of recycling) of mixing only polymeric matrix and wood fibers without additives or additional fiber treatment.

Thus, the aim of this research is twofold. First, to predict and compare WPCs’ linear viscoelastic long-term creep behavior based on short-term creep experiments using recycled (RPP) and non-recycled/neat (NPP) polypropylene matrix at different fiber concentrations. The second goal of this research is to show that adding wood fibers to the recycled (RPP) PP matrix leads to as good or superior long-term creep properties compared to a non-recycled/neat (NPP) polypropylene matrix. As recycled polymeric materials usually exhibit worse mechanical properties than neat polymers, wood fibers could be effectively used to improve the creep properties of recycled materials.

## 2. Materials and Methods

### 2.1. Materials and Sample Preparation

For the investigation, two polypropylene matrix materials were used. Non-recycled or neat Amppleo 1025MA Polypropylene (NPP), Braskem, Rotterdam, The Netherlands, and recycled Eco Meplen IC M20 BK (RPP), Mepol S.r., Treviso, Italy. Both materials were filled with wood fibers at different loading levels without additives. The fibers consisted of spruce and pine wood (ratio approx. 80–20%) with a mean volume equivalent sphere fiber diameter of 99.58 μm determined by the laser diffraction method. Material mixtures were prepared by extrusion. The final dumbbell samples were injection molded (sample type 1B, ISO 3167:2014). Prior to injection molding, granules were dried at 80 °C. The injection molding temperature was 190 °C and the cooling time in the mold was 12 s. The dimensions of the dumbbell samples were roughly 75 mm × 10 mm × 2 mm.

The complete materials and samples preparation procedure is described elsewhere [36,37]. The overview of investigated materials is given in Table 1.

### 2.2. Thermal Characterization

The thermal properties of the materials were determined with a differential scanning calorimeter, TA DSC2500, TA Instruments, New Castle, DE, USA. The long and flat side of the dumbbell samples was used to punch out DSC samples. About 10 mg samples were prepared from the middle part of dumbbell samples. A heat-cool-heat temperature scan was performed with a heating/cooling rate of 10 °C/min in the N_2_ atmosphere (50 mL/min). Tests were conducted from −70 °C to 200 °C. For each material, at least three repetitions were made. In the present study, only first heating was used to analyze the degree of crystallinity (χ) and melting temperature (*T_m_*) to obtain the thermal properties of samples that were used for creep tests. The degree of crystallinity was determined following Equation (1):(1)$$\chi =\frac{∆H_{m}}{∆H_{m}^{0}\cdot \left(1-W_{wt.\%}\right)}\times 100$$
where χ is the degree of crystallinity, $∆H_{m}$ is the melting enthalpy, $∆H_{m}^{0}$ is the melting enthalpy of the 100% crystalline PP and is 207 J/g [38]. $W_{wt.\%}$ is the mass fraction of wood fibers in the material.

### 2.3. High-Temperature Gel Permeation Chromatography

The molecular weight distribution (MWD), number average molecular weight (M_n_), weight average molecular weight (M_w_), and polydispersity (M_w_/M_n_) of both matrix materials were determined using the high-temperature gel permeation chromatography (H-GPC), GPC—IR5 Polymer Char, Polymer Char, Valencia, Spain. The measurements were done at 160 °C using three 300 × 7.5 mm PL gel Olexis Mix-Bed columns (13 μm). Materials were dissolved in trichlorobenzene (TCB) and stabilized with butylhydroxytoluene (BHT) for 90 min. The flow rate of the mobile phase was 1 mL/min. For calibration, a narrow polystyrene standard was used.

### 2.4. Creep Compliance, Long-Term Creep Properties, and Mechanical Spectrum

Shear creep experiments have been performed with a rotational rheometer, MCR 302 Anton Paar, Graz, Austria, on injection molded samples. Prior to measuring, the samples were tempered at 90 °C for 3 h. Tempering was followed by gradual cooling (0.1 °C/min) to the ambient temperature. Short-term creep compliance segments were determined by applying step shear stress, τ = 0.01 MPa for 1000 s in the temperature range from −10 °C to 80 °C. The shear stress level was selected to be in the linear viscoelastic range. A single sample was used to measure creep compliance at all temperatures, and at least three repetitions were made per material. The measuring procedure was done following these steps:Once the sample was at the ambient temperature, it was clamped into the rheometer and cooled (0.1 °C/min) to the first measuring temperature of −10 °C.The measuring procedure consisted of 1 h temperature stabilization, followed by application of constant shear stress for 1000 s.After each loading, samples were unloaded, and the temperature was raised for 10 °C.During the complete measuring procedure in the rheometer, the normal force was set to 0 N to prevent normal stresses in the sample due to thermal expansion.

The creep compliance of materials was calculated using Equation (2):(2)$$J\left(t\right)=\frac{\gamma \left(t\right)}{\tau_{0}}$$
where the γ(t) is the measured shear strain and the $\tau_{0}$ is the applied constant shear stress.

To predict long-term creep compliance, time-temperature superposition (tTSP) was used to construct the mastercurves from the individual segments measured at different temperatures. In the present study, the Closed-Form Shifting algorithm [39,40] was used to construct the unique mastercurves at a selected reference temperature $T_{ref}=20\;{}^{\circ}C$. Smooth mastercurves with the same number of data points for each material were obtained using a polynomial fit. Details on the data smoothing procedure are provided in Section A.2. Parameters for smoothing the mastercurves, Equation (A1) and Table A1.

WPC material creep behavior can be described with rheological models or mechanical spectrum [41]. In this case, creep compliance is represented as follows:(3)$$J\left(t\right)=J_{g}+\overset{n-1}{\underset{i=1}{\sum }}J_{i}\left(1-exp\left(-t/\tau_{i}\right)\right)+\frac{t}{\eta_{0}},$$
where *J_g_* is glassy (i.e., instantaneous) creep compliance, i is an index of Voigt element, *n* represents a number of elements with retardation times $\tau_{i}$ and intensity values *J_i_*, t is time, and $\eta_{0}$ is a terminal velocity. The last term of Equation (3), which is also referred to as the flow term, is neglected in the present study. With a sufficient number of elements, parameters *J_i_* and $\tau_{i}$ represent retardation mechanical spectrum of the material L(τ). Using a finite number of Voigt elements results in a discrete retardation spectrum. The intensity of each spectral line represents the individual contribution to the total material response, and response times can be related to the molecular size/weight [41].

Within this investigation, an assessment of the mechanical spectrum was done using open source software RepTate [42]. Fourteen nodes were used for spectrum generation. Nodes are distributed equally in the logarithmic time scale, and the fitting procedure is performed using Trust Region Reflective algorithm.

### 2.5. Scanning Electron Microscopy

Scanning electron microscopy (SEM) was used to evaluate the polymer–fiber interface on the fractured surface of composite materials. First, samples were dipped in liquid nitrogen for about 3 min. Afterward, they were fractured and sputter-coated with a 20 nm thick gold coating (sputtering at an mm working distance using 30 mA for 100 s) using SCD005, Baltec AG, Wetter/Ruhr, Germany. SEM analysis was performed using JEOL JSM-163 IT100, Tokyo, Japan at an accelerating voltage of 20 kV.

### 2.6. X-ray Diffractometry

The identification of crystalline phases and orientation of thermoplastic composites filled with wood fibers were analyzed with X-ray diffractometry (XRD). The XRD spectra were taken using a Siemens D5000, Munich, Germany diffractometer using Cu K α radiation source at 40 kV. The X-ray diffraction patterns were recorded for the angles in the range of 2θ, with a step of 3°/min (λ = 0.154) and measured from 5 to 30°. The spectra were fitted using the crystallographic program Topas2R 2000 (Bruker AXS, Billerica, MS, USA) based on a convolution approach (Pawley method).

## 3. Results and Discussion

### 3.1. Molecular Structure of NPP and RPP Matrix Materials

Figure 1 shows the molecular weight distribution of recycled (RPP) and neat (NPP) polypropylene matrix. The differences between materials are small. However, a clear shift of molecular weights to smaller sizes in the case of RPP material can be observed. In addition, the number and weight average molecular weights (M_w_ and M_n_) of RPP material are lower than NPP material (Table 2). However, the polydispersity index (M_w_/M_n_) of RPP is higher than NPP material.

### 3.2. Thermal Properties

The crystallinity of polymers has a vital role in determining their mechanical properties. It is generally known that higher crystallinity of material results in higher creep resistance, elastic modulus, and lower toughness and strength. Table 3 shows the melting temperature, $T_{m}$, melting enthalpy, ∆*H_m_*, and degree of crystallinity, χ, of composites with NPP and RPP matrix. The thermal properties correspond to the properties of the samples used for creep tests as the first heating was used for the analysis.

Results show that the melting temperature, *T_m_*, does not significantly change regardless of the polymeric matrix or wood fiber concentration. With the increase of wood fibers, the crystallinity of injection molded samples increases. Similar behavior was also observed for granulated composite materials used to prepare these samples [36]. In addition, the presence of wood fibers promotes the nucleation process as the wood surface provides a large number of potential nucleation sites [32].

The crystallinity of composites with NPP matrix increased from 41.9% to 53.9%, corresponding to a 22% increase. On the other hand, the crystallinity of composites with RPP matrix, which exhibit lower crystallinity than NPP matrix composites, increased from 33.7% to 45.6%, corresponding to a 26% increase. The difference in the crystallinity between NPP and RPP matrices may be attributed to small differences in the MWD. Evaluating the absolute difference in crystallinity between RPP and NPP matrix composites at a specific wood fiber concentration:(4)$$\Delta =\left|\chi_{NPP}-\chi_{RPP}\right|$$
shows no trend as the difference in crystallinity is between 6.8 and 8.3% (Table 3).

### 3.3. Long-Term Creep Prediction and Mechanical Spectrum

Long-term creep compliance mastercurves of WPC composites were constructed using time-temperature superposition. The creep compliance mastercurves for composites with NPP matrix are shown in Figure 2a and composites with RPP matrix in Figure 2b for *T_ref_* = 20 °C. The full lines represent the shifted raw mastercurves, while symbols represent the smooth mastercurve. The maximal relative error of all creep compliance measurements is between 4 and 9%. The individual segments with error bars, corresponding mastercurves, and shift factors are presented in Figure A1 and Figure A2.

As evident from Figure 2a,b, a generally known trend can be observed—the addition of wood fibers reduces creep compliance. For example, creep compliance of composite with 40 wt.% compared to the compliance of matrix material (0 wt.%) is lower by about 40% after 1 year (dash-dot line) for both NPP and RPP matrix materials.

A direct comparison between creep compliance of composites with NPP and RPP matrix is shown in Figure 2c,d. For clarity, only the compliance of matrix materials (0 wt.%) and composites with 20 and 40 wt.% are shown in Figure 2c. The comparison shows that the creep compliance of composites with the RPP matrix is higher than composites with the NPP matrix at all concentrations. This is in line with the measured higher degree of crystallinity of composites with NPP matrix, as the higher crystallinity causes lower creep compliance [43,44].

However, the difference in creep behavior between the composites with two different matrix materials changes with time and wood fiber concentration. This is vividly shown in Figure 2d, where the relative difference in creep compliance between composites with NPP and RPP matrix is shown. This comparison shows that the difference between the two matrix materials is generally smaller as the wood fiber concentration increases. The decreasing amount of polymer in composite could explain this since there is less material in the composite that creeps. Thus, the overall creep response becomes less dependent on the matrix material. For all wood fiber concentrations, it can also be seen that the difference between the creep is increasing with time. This means that the composites with RPP matrix creep more than those with NPP matrix. At longer times, the difference increases faster (marked with a dotted line in Figure 2d).

The increase of creep relative difference does not apply to the matrix material (0 vol.%). However, in the whole timescale, the difference between NPP and RPP matrix materials (at 0 vol.%) changes from 20% (at short times) to 15% (at longer times). This indicates that the fibers affect the creep process differently if NPP or RPP matrix is used.

As no additives were used to prepare composites, the interfacial adhesion is expected to be poor. However, there is still a difference in creep between composites with two matrices. Therefore, to investigate the fiber-polymer interaction more closely and to gain a better insight into the creep mechanisms, the molecular processes related to the time-dependent properties and the effect of fiber concentration were investigated through the retardation spectrum. The spectra of composites with NPP matrix are shown in Figure 3a, and the spectra of composites with RPP matrix in Figure 3b.

For both matrix materials, the spectrum intensity decreases with wood fiber concentration. However, the shape of the spectra is generally retained. The spectra also reveal less intensive molecular mobility at longer times when wood fibers are added, as the spectrum intensity decreases. Simultaneously, the impact of wood fibers at shorter times is smaller as the differences in spectrum intensity between the neat matrix material and filled materials are smaller. The result implies that the molecular motion is less restricted by wood fibers for shorter molecular chains (corresponding to shorter times), while the movement of longer chains (corresponding to longer times) is hindered more by wood fibers, as seen by the decreased spectra intensity.

For a closer comparison of the spectra shape, Figure 3c,d show a normalized spectrum of composites with NPP and RPP matrix. Each spectral line was normalized to the sum of all spectra lines, Equation (5):(5)$$l_{i}\left(\tau_{i}\right)=\frac{L_{i}\left(\tau_{i}\right)}{\sum_{j=1}^{14}L_{j}},\;i=1,2,\ldots ,14$$
where *L_i_* represents the weight of the retardation spectrum line at a given time, *τ* is time, and *L_j_* is the sum of all spectral lines.

The retention of the normalized spectra shape, regardless of the wood fiber concentration, leads to the conclusion that there are no direct interactions between fibers and polymeric matrix on a molecular scale that would affect the time-dependent response of composite materials. Furthermore, as mentioned, no additives were used to prepare composite materials, so it is also unlikely that there would be any indirect interaction between fibers and polymer through the boundary layer formed by additives, which are chemical bridges between fibers and the polymeric matrix. The ATR-FTIR analysis shown in the previous publication confirmed the absence of fiber-matrix chemical interactions [36].

Thus, the decreasing values of spectra and reduced creep compliance with increasing wood fiber concentration (Figure 3a,b) may be attributed to the reduced relative volume of the matrix polymer in the composite. Additionally, wood fibers also affect the composite time-dependent response as part of the stress is locally sustained by relatively rigid fibers. Furthermore, a 3D-connected fiber network is formed at a high enough wood fiber concentration, which can transfer load more efficiently and reduce creep further through fiber-to-fiber load transfer. However, these two mechanisms still cannot explain the more severe creep of composites with RPP matrix at longer times shown in Figure 2d.

### 3.4. SEM Images

To further investigate the influence of matrix material on the fiber-polymer interaction and creep compliance, SEM micrographs of the fractured surfaces are shown in Figure 4. Figure 4a shows a composite with an NPP matrix at 20 wt.% several voids reflecting the fiber shape that can be seen (arrows). These were caused by fibers pulled out during sample fracture indicating weakly bonded fibers to the matrix. However, it may also be seen that no gap between fibers and polymer occurs in some instances (circle), indicating good interfacial adhesion. In the magnified image shown in Figure 4c, a good fiber–polymer interaction (absence of a gap between them) can be seen even more clearly (arrow). Additionally, a fiber failure (dotted arrow) can also be seen that confirms good adhesion.

The analysis of SEM images demonstrates that the composites with RPP matrix form a weaker fiber–polymer interface than composites with NPP matrix. The same observations can also be made for other wood concentrations. These images are shown in Figure A3.

The fracture surface of a composite with RPP matrix and 20 wt.% of wood fibers in Figure 4b shows a different situation as the fibers are more exposed than the composites with NPP matrix. Several voids caused by fiber pullout (arrows) may also be seen. In contrast to Figure 4a, practically all fibers are separated from the RPP matrix. In the magnified image in Figure 4d, a large gap between the fibers and matrix can be seen.

### 3.5. XRD Measurements

The fiber–polymer interactions and differences between the composites with NPP and RPP matrix were further investigated by X-ray diffraction (XRD). The monoclinic α-phase is one of the most common polymorphic structures observed for polypropylene material as it is the most thermodynamically stable [45,46]. However, it is known that (natural) fillers can cause changes in the crystallinity and morphology of polypropylene [47,48,49,50,51,52,53]. When the composite material is injected into a mold, a combination of high local shear forces and rapid cooling in the mold cause several nucleation sites at the fiber surface, which inhibit lateral crystal growth and thus take place only in a normal direction to the fiber [53]. The growth of this oriented crystal layer (transcrystalline layer) is limited by the spherulite growth in the bulk polymer [54,55]. It is well known that the transcrystalline phase around fibers has a very positive effect on the interfacial adhesion, load transfer, and in turn, on (thermo) mechanical properties [33,34,35].

For granules used to make samples for this investigation, hexagonal β modification and γ polymorphic structures in PP-based materials were already observed [36]. However, in continuation, the XRD results on the (injection-molded) samples used to measure creep properties are presented.

The X-ray diffraction patterns in the 2θ angle range of the NPP matrix composites as a function of wood fiber concentration are shown in Figure 5 at 14.04° (110), 16.07° (300), 16.78° (004), 18.53° (130), 21.08° (311), 21.70° (−131), 25.43° (0012), and 28.47° (006). In the diffraction pattern, peaks of the α-phase, β-phase, and γ-phase can be distinguished for all materials. The reflection peaks of composite materials at 14.04°, 16.78°, 18.53°, 21.7°, and 28.4° reflect the monoclinic α form, while the peaks at 16.07° and 21.08° indicate the β-phase and at the 25.43° also the γ-phase.

The XRD results show that the matrix NPP material (0 wt.%) is isotactic, where the chains are packed into the lattice as a left- or right-handed 2*3_1_ helical conformation with methyl groups facing up or down [46,56]. This tacticity enables the formation of polymorphic α, β, and γ-phases.

XRD results show an increased peak intensity at the diffraction pattern peaks of the β-phase (at 16.07 and 21.08) with increasing fiber concentration. When the material is injection-molded, sufficiently large local stresses at the fiber–polymer phase trigger the formation of a transcrystalline layer (shear-induced crystallization). In [57], it was argued that the dominant role in the formation of transcrystalline layer is mainly related to fiber surface roughness. Fibers used in this study have a very irregular and quite rough surface (Figure 6), with several fibrils extending from the main fiber, which promotes the formation of transcrystallization. As more fibers (surfaces) are available, nucleation is also more likely to occur. Therefore, an increased β-phase intensity peak could be associated with the wood fiber concentration and possibly also the difference in the crystallite size of the (300) and (311) planes.

Figure 7 shows the XRD diffraction patterns in the 2θ angle range of the RPP matrix composites as a function of wood fiber concentration. The diffraction peaks of composites with RPP matrix are positioned at 2θ angles of 9.38° (200), 14.04° (110), 16.87° (040), 18.52° (130), 21.84° (−131), 25.43° (0012), 28.61° (411), and 29.40° (3.03).

Results show two tacticities in the composites with RPP matrix; isotactic and syndiotactic. The diffraction peaks of composites with the RPP matrix do not show any evidence of the formation of β-phase. Instead, the peaks mainly reflect the α-phase of polypropylene. Additionally, the wood fiber concentration does not affect the polymorphic crystalline structure, as this is the case for composites with NPP matrix. Furthermore, wood fibers do not act as nucleating agents, as is the case for composites with NPP matrices. Thus, the peak intensity decreases with higher wood fiber concentration.

From the presented results, it can be concluded that the better polymer–fiber adhesion (Figure 4a,c) can be attributed to the transcrystalline morphology of composites with NPP matrix. Stronger polymer–fiber adhesion could also explain the differences in the creep behavior between composites with NPP and RPP matrix. Namely, the slow creep of composites with NPP matrix at longer times (Figure 2d) is caused by the transcrystalline morphology. Results also show that composites with RPP matrix do not form the transcrystalline layer, although the same wood fibers and sample preparation procedure were used. This could be attributed to the two tacticities detected by XRD in the RPP matrix (isotactic and syndiotactic). One of the possible factors affecting the occurrence of transcrystallization could also be the molecular weight. Generally, higher molecular weight lowers the required local stress that triggers the shear-induced crystallization [58]. As shown in Figure 1 and Table 2, the NPP matrix material has a higher molecular weight than the RPP matrix material. However, the difference in molecular weight, M_w_, is relatively small, about 10%. Nonetheless, both the difference in tacticity and molecular weight likely promote or suppress the transcrystallization.

### 3.6. Improving Creep Properties of the Recycled Matrix Using Wood Fibers

As recycled polymers usually have inferior mechanical properties to non-recycled/neat polymers, fillers can be a good way to increase these properties. As shown in Figure 2, using untreated wood fibers can substantially decrease creep compliance. The results also showed that composites with RPP matrix exhibit higher creep compliance than composites with NPP matrix as they do not form transcrystalline morphology. Regardless, introducing untreated fibers to the recycled polymeric matrix can still be beneficial to decrease its long-term creep compliance and even surpass the creep performance of the non-recycled/neat polymers. To this end, a relative comparison between creep compliance of composites with RPP matrix at particular wood fiber concentration to non-recycled/neat (NPP) matrix was made:(6)$$\delta \left(t,x\right)=\left(1-\frac{J_{RPP,\;x\;wt.\%\;}\left(t\right)}{J_{NPP,\;\;0\;wt.\%}\left(t\right)}\right)\;\times 100\%$$

$J_{RPP,\;x\;wt.\%}$ is creep compliance of composites at a given wood fiber concentration, x wt.% is particular wood fiber concentration and the $J_{NPP,\;0\;wt.\%}$ is the creep compliance of non-filled (0 wt.%) neat matrix (NPP). The results of this analysis are shown in Figure 8.

The analysis confirms previous findings that the RPP matrix (0 wt.%) creeps more than the NPP matrix (dashed line). The difference between both matrix materials varies with time but is between 20–30% in the whole measured time. As expected, by increasing the wood fiber concentration, the difference between the composites with RPP and NPP matrix starts to decrease. At 10 wt.% and longer times, the creep compliance of composite is comparable to the creep of NPP matrix material. As the concentration of 10 wt.% is exceeded, the increased creep performances can be seen for the whole time scale. Adding 20 wt.% of untreated fibers to the RPP matrix decreases creep compliance compared to the NPP matrix for 10–20%, while adding 40 wt.% causes more than 50% creep improvement.

## 4. Conclusions

This study has shown that untreated wood fibers decrease long-term creep compliance of materials, regardless of the polymeric matrix. Higher wood fiber concentrations lead to more significant creep compliance reduction as there is less polymeric material that creeps. The difference in creep behavior between recycled (RPP) and neat (NPP) matrix materials can be partially connected to the difference in their degree of crystallinity. The composites with RPP matrix show 6–8% lower crystallinity than composites with NPP matrix.

Furthermore, a transcrystalline layer between fibers and bulk polymer for the composites with NPP matrix was detected with XRD measurements. On the other hand, transcrystallinity was not observed for any composites with the RPP matrix. We argue that this difference in morphology causes better fiber–polymer adhesion observed with SEM and better long-term creep stability of composites with NPP matrix. The transcrystallinity in an NPP matrix is caused by processing conditions and isotactic morphology of the matrix polypropylene. In addition, the rough wood fibers’ surface acts as a nucleating agent for crystallization and further promotes the formation of transcrystallinity. The paper demonstrated that long-term creep compliance of composites with NPP matrix could be decreased up to 25% compared to creep compliance of composites with recycled (RPP) polypropylene matrix, which does not form transcrystalline layer between the fibers and polymer matrix.

Additionally, it was demonstrated that adding wood fibers to the recycled matrix (RPP) can improve the creep properties by up to 50% compared to the creep properties of the non-recycled/neat (NPP) matrix. While such composites may not provide the same creep properties as wood–polymer composites with high interfacial adhesion between fibers and polymer, they can still be used to decrease the creep compliance of recycled material. As shown, the creep compliance of composites with recycled (RPP) matrix can even surpass the performance of recycled/neat (NPP) matrix. Thus, using recycled matrix material and simply mixing it with wood fibers could be a strong tool for improving the creep performance of materials and increasing sustainability and the use of waste/recycled materials.

In conclusion, using untreated wood fibers in wood polymer composites has some obvious disadvantages. Incompatibility between the hydrophobic polymeric matrix and hydrophilic wood fibers results in bad interfacial adhesion, which results in high creep compliance and decreased durability. In addition, a weak interface can potentially facilitate the access of humidity into a composite. However, this research showed that substantial creep reductions can still be achieved by adding untreated wood fibers to the polymeric matrix. In this respect, composites with untreated wood fibers might be useful in cases where fiber treatment is economically unsustainable or environmentally more acceptable, and stable environmental conditions are provided.

## Figures and Tables

**Figure 1 polymers-14-02539-f001:**
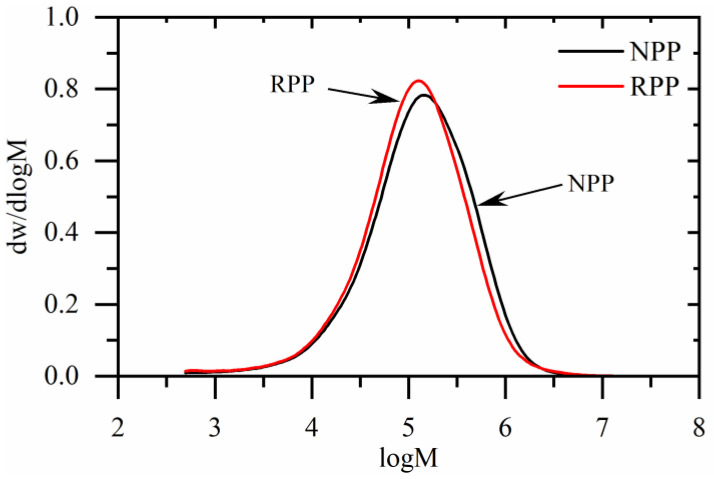
The molecular weight distribution of neat (NPP) and recycled (RPP) polypropylene matrix materials.

**Figure 2 polymers-14-02539-f002:**
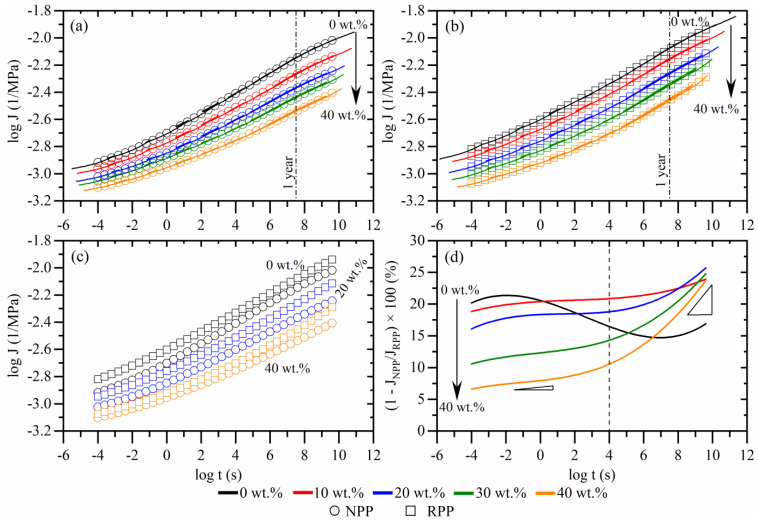
Shear creep compliance mastercurves at *T_ref_* = 20 °C of composites with neat (NPP) matrix (**a**) and recycled (RPP) matrix (**b**); (**c**) comparison of creep with NPP and RPP matrix at selected wood fiber concentrations; (**d**) relative difference of creep compliance between composites with NPP and RPP matrix.

**Figure 3 polymers-14-02539-f003:**
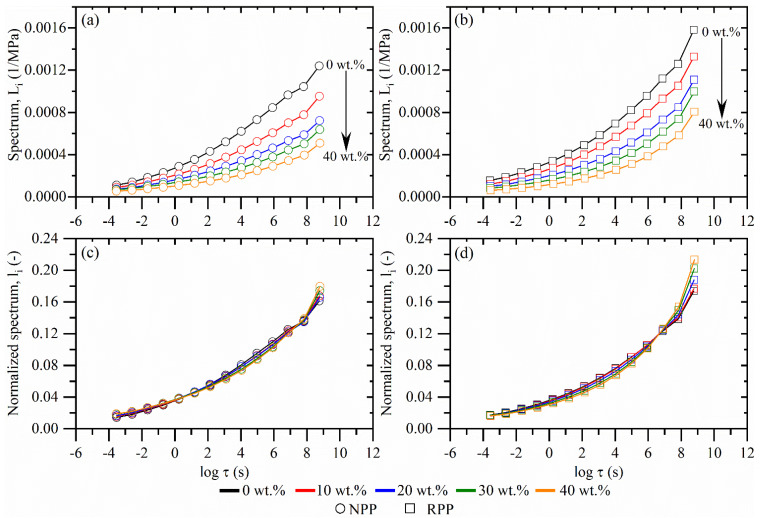
Retardation spectrum of composites with (**a**) neat (NPP) and (**b**) recycled (RPP) matrix. Normalized retardation spectrum of composites with (**c**) neat (NPP) and (**d**) recycled (RPP) matrix.

**Figure 4 polymers-14-02539-f004:**
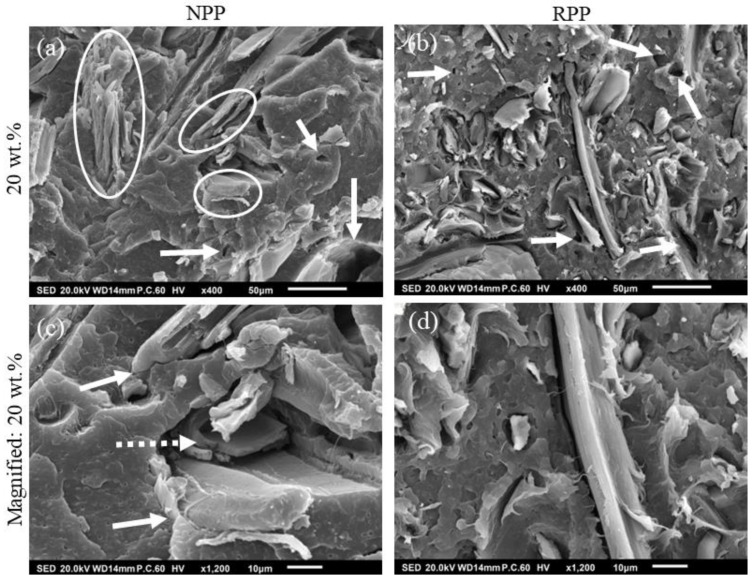
SEM images of fractured samples with 20 wt.% wood fibers; (**a**) neat (NPP) and (**b**) recycled (RPP) matrix and magnified images of (**c**) neat (NPP) and (**d**) recycled (RPP) matrix.

**Figure 5 polymers-14-02539-f005:**
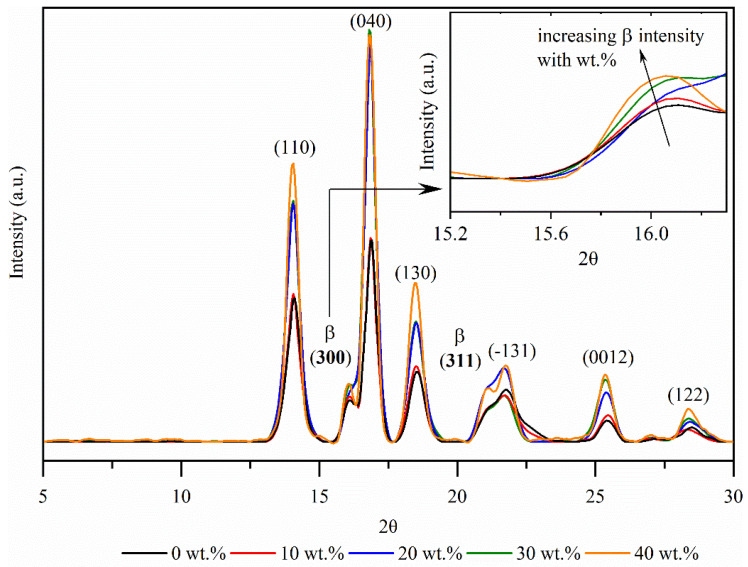
X-ray diffraction (XRD) spectra of the composites with a neat (NPP) matrix.

**Figure 6 polymers-14-02539-f006:**
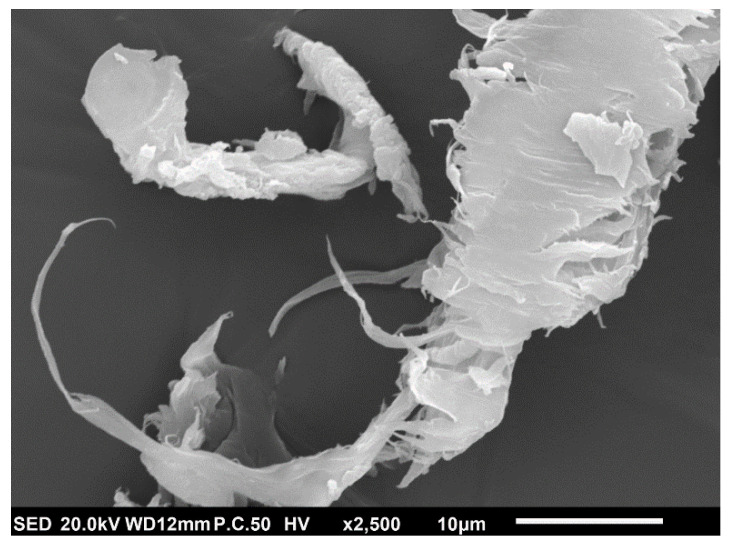
SEM image showing the surface of the wood fiber.

**Figure 7 polymers-14-02539-f007:**
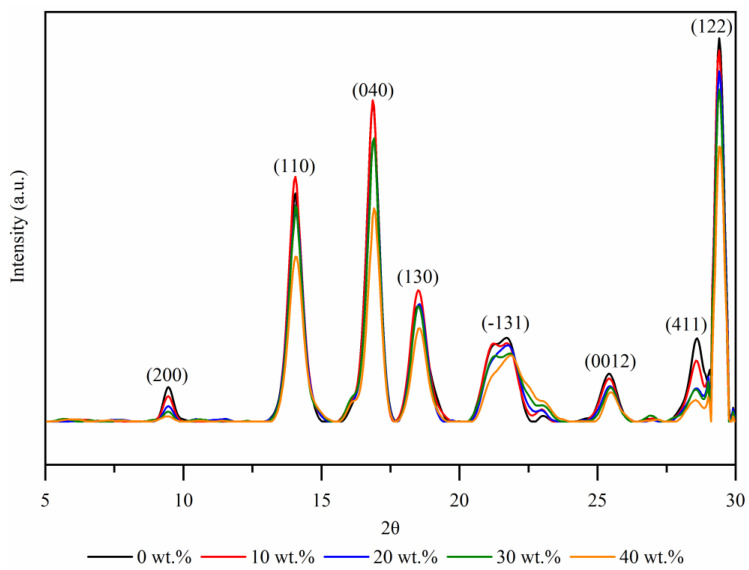
X-ray diffraction (XRD) spectra of the composites with neat (RPP) matrix.

**Figure 8 polymers-14-02539-f008:**
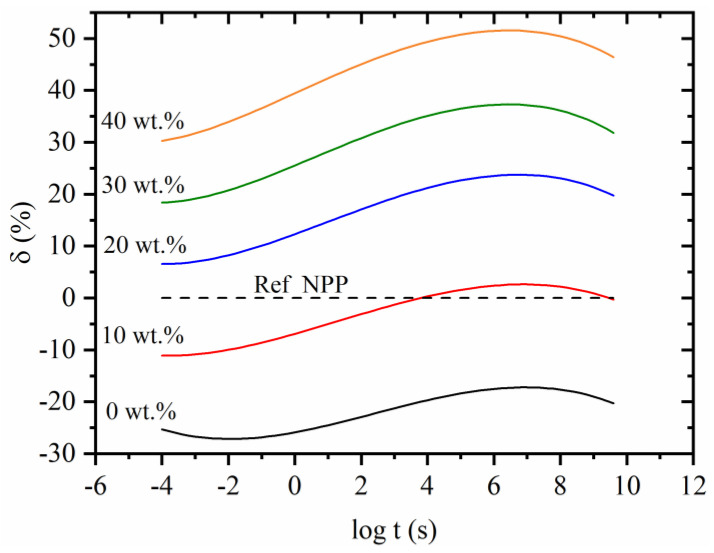
Creep compliance of composites with recycled (RPP) matrix relative to neat (NPP) matrix at $T_{ref}=20\;{}^{\circ}C$.

**Table 1 polymers-14-02539-t001:** Nomenclature and sample material composition.

Polymer Matrix	Abbreviation	Wood Fibers Concentration (wt.%)				
Neat Polypropylene (Amppleo 1025MA)	NPP	0	10	20	30	40
Recycled Polypropylene (Eco Meplen IC M20 BK)	RPP					

**Table 2 polymers-14-02539-t002:** High-temperature gel permeation chromatography (H-GPC) results.

Polymer Matrix	M_w_ (g/mol)	M_n_ (g/mol)	M_w_/M_n_ (-)
NPP	246,880	38,930	6.3
RPP	221,910	33,750	6.6

**Table 3 polymers-14-02539-t003:** Thermal characteristics of composites with neat (NPP) and recycled (RPP) polypropylene matrix.

Wood Fibers (wt.%)	NPP Matrix			RPP Matrix			Δ (%)
	$T_{m,PP\;}\;({}^{\circ}C)$	$∆H_{m,PP}\;\;(J/g)$	$\chi_{PP\;}\;(\%)$	$T_{m,RPP}\;\;({}^{\circ}C)$	$∆H_{m,RPP\;}\;(J/g)$	$\chi_{RPP}\;\;(\%)$	
0	165.1 ± 0.1	86.8 ± 0.5	41.9 ± 0.2	166.0 ± 0.3	69.8 ± 0.8	33.7 ± 0.4	8.2
10	165.5 ± 0.6	82.7 ± 1.9	44.4 ± 1.0	165.6 ± 0.2	68.7 ± 1.2	36.9 ± 0.7	7.5
20	165.5 ± 0.3	76.6 ± 2.7	46.3 ± 1.7	165.2 ± 0.4	65.1 ± 1.9	39.3 ± 1.2	7.0
30	165.4 ± 0.2	73.0 ± 1.8	50.4 ± 1.2	166.1 ± 0.2	63.2 ± 2.1	43.6 ± 1.4	6.8
40	165.1 ± 0.2	67.0 ± 3.0	53.9 ± 2.4	164.9 ± 0.8	56.6 ± 7.8	45.6 ± 6.3	8.3

## Data Availability

The data presented in this study are available on request from the corresponding author.

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
