# Peer review of "Long-Term Creep Compliance of Wood Polymer Composites: Using Untreated Wood Fibers as a Filler in Recycled and Neat Polypropylene Matrix"

_polymers, 2022, doi:10.3390/polym14132539_

Round 1

Reviewer 1 Report

The paper presents the results of research on WPC composites containing up to 40% wood flour based on neat and PP recyclate.
The authors characterized the molecular weight of selected matrices and carried out tests of the produced composites using the DSC, SEM, XRD methods. They also rated the long-term creep properties and mechanical spectrum of obtained composites.
In my opinion, it is an interesting and interesting job. The authors also noted the novelty of the approach to the problem of creep of WPC composites with unmodified filler.
The manuscript was thoroughly prepared and the conclusions were supported by the results of the work. In my opinion, the manuscript requires a few minor additions:
Despite references to authors' other works, please include in the manuscript the conditions for the production of WPC composites along with the temperature of the injection mold and cooling time and the dimensional characteristics of the sample obtained.

Was the obtained WPC granulate additionally dried before injection?

How were samples taken for DSC testing? from which part of the bone-like sample? Did the selected sample for DSC test cover the entire cross-section of the bone-like sample? How many repetitions of DSC measurement were made? Why the DSC curves obtained during cooling were not analyzed?

What was the form of the sample for XRD testing? With this type of injection fittings, a crystallinity gradient may develop depending on the sample cross-section (the same remark applies to DSC tests).

Reviewer 2 Report

This research is about the use of wood in polymer composites, especially the fiber as fillers for PP recycling. 

1. where is the laser diffraction data for the wood fiber distributions? 

2. composite mechanics-wise, can the authors also provide the estimation about the mechanical behaviors in tension or compression?
3. what is the dispersion quality of wood fibers at higher concentrations? beyond 20 wt%?
